Ropivacaine inhibits the malignant behavior of lung cancer cells by regulating retinoblastoma-binding protein 4

Jia Weiai 1
Shen Junmei 1
Wei Sisi 2
Li Chao 1
Shi Jingpu 1
Zhao Lianmei 2
Jia Huiqun jysyjiahuiqun@163.com 1
1 Department of Anesthesiology, The Fourth Hospital of Hebei Medical University , Shijiazhuang , Hebei , China
2 Scientific Research Center, The Forth Hospital of Hebei Medical University , Shijiazhuang , Hebei , China
Tyagi Abhishek
Electronic publication date: 2023 Nov 27
Publication date: 2023
Volume: 11
Electronic Location ID: e16471
Received 2023 Jun 26; Accepted 2023 Oct 25
Copyright: ©2023 Jia et al.
Copyright year: 2023
Copyright holder: Jia et al.
License: This is an open access article distributed under the terms of the Creative Commons Attribution License, which permits unrestricted use, distribution, reproduction and adaptation in any medium and for any purpose provided that it is properly attributed. For attribution, the original author(s), title, publication source (PeerJ) and either DOI or URL of the article must be cited.
License URL: https://creativecommons.org/licenses/by/4.0/

Keywords: Ropivacaine, Lung cancer, RBBP4, Proteomics, Malignant behavior

Funding: The authors received no funding for this work.

==============================
Background

Ropivacaine is a local anesthetic commonly used in regional nerve blocks to manage perioperative pain during lung cancer surgery. Recently, the antitumor potential of ropivacaine has received considerable attention. Our previous study showed that ropivacaine treatment inhibits the malignant behavior of lung cancer cells in vitro. However, the potential targets of ropivacaine in lung cancer cells have not yet been fully identified. This study aimed to explore the antitumor effects and mechanisms of action of ropivacaine in lung cancer.

Methods

Lung cancer A549 cells were treated with or without 1 mM ropivacaine for 48 h. Quantitative proteomics was performed to identify the differentially expressed proteins (DEPs) triggered by ropivacaine treatment. STRING and Cytoscape were used to construct protein-protein interaction (PPI) networks and analyze the most significant hub genes. Overexpression plasmids and small interfering RNA were used to modulate the expression of key DEPs in A549 and H1299 cells. MTS, transwell assays, and flow cytometry were performed to determine whether the key DEPs were closely related to the anticancer effect of ropivacaine on the malignant behavior of A549 and H1299 cells.

Results

Quantitative proteomic analysis identified 327 DEPs (185 upregulated and 142 downregulated proteins) following ropivacaine treatment. Retinoblastoma-binding protein 4 (RBBP4) was one of the downregulated DEPs and was selected as the hub protein. TCGA database showed that RBBP4 was significantly upregulated in lung cancer and was associated with poor patient prognosis. Inhibition of RBBP4 by siRNA resulted in a significant decrease in the proliferation and invasive capacity of lung cancer cells and the induction of cell cycle arrest. Additionally, the results indicated RBBP4 knockdown enhanced antitumor effect of ropivacaine on A549 and H1299 cells. Conversely, the overexpression of RBBP4 using plasmids reversed the inhibitory effects of ropivacaine.

Conclusion

Our data suggest that ropivacaine suppresses lung cancer cell malignancy by downregulating RBBP4 protein expression, which may help clarify the mechanisms underlying the antitumor effects of ropivacaine.

Introduction

Lung cancer is a widespread malignancy and the leading cause of cancer-related fatalities worldwide (Miller et al., 2022; Zheng et al., 2022). According to global cancer statistics, an estimated 2.2 million new cases and 1.8 million deaths from lung cancer were recorded in 2020 (Sung et al., 2021). Surgical resection remains the primary treatment for early-stage lung cancer. However, postoperative recurrence and metastasis are the main factors contributing to the low survival rates of patients with lung cancer (Ng, Zhao & Lau, 2017). Perioperative management, including the use of local anesthetics during surgery, may affect lung cancer outcomes (Cata et al., 2014; Lee et al., 2017). Most current studies support the idea that intravenous propofol is more advantageous than volatile inhalational anesthetics in reducing cancer recurrence (Jansen, Dubois & Hollmann, 2022). A preclinical study showed that propofol treatment inhibited cell growth and accelerated apoptosis of lung cancer A549 cells by regulating the miR-21/PTEN/AKT pathway in vitro (Zheng et al., 2020). In a retrospective clinical study, propofol-based total intravenous anesthesia was reported to have better long-term oncologic outcomes than inhalation anesthesia in patients with lung cancer who underwent curative resection (Seo et al., 2022). Moreover, local anesthetics are believed to suppress surgical stress, decrease opioid consumption, and preserve cancer-related immune function, potentially improving the prognosis of patients with cancer.

Ropivacaine is one of the most commonly used amide-linked local anesthetics for lung cancer surgery. Thoracic epidural analgesia or other nerve blocks using ropivacaine are of great importance in perioperative pain management (Tamura et al., 2017). Additionally, recent studies have shown that ropivacaine has antitumor effects in various cancers, such as lung cancer (Wang et al., 2016; Piegeler et al., 2015), hepatocellular carcinoma (Chen et al., 2020; Zhang et al., 2021), colon cancer (Baptista-Hon et al., 2014; Wang & Li, 2021), gastric cancer, (Zhang et al., 2020) and pancreatic cancer (Bundscherer et al., 2015) by multiple molecular mechanisms. A preclinical study demonstrated that ropivacaine at clinically relevant concentrations of 1 nM-100 µM could significantly inhibit lung adenocarcinoma cells invasion and matrix-metalloproteinases-9 secretion by blocking the activation of Akt and focal adhesion kinase (Piegeler et al., 2015). Our previous study showed that ropivacaine inhibits the invasive and metastatic ability of A549 and H1299 lung cancer cells, potentially by regulating the expression of hypoxia inducible factor (HIF) 1 α, Vascular endothelial growth factor (VEGF), and matrix metalloproteinase (MMPs) (Shen et al., 2022). However, the potential pharmacological targets of ropivacaine in lung cancer cells have not yet been fully elucidated.

In recent years, high-throughput mass spectrometry-based proteomics has become a commonly used technique for protein identification and quantification (Aebersold & Mann, 2016). It is also widely used to screen for differentially expressed proteins (DEPs), tumor markers, and prognostic markers in various tumors, including lung cancers (Gillette et al., 2020). In the present study, high-performance liquid chromatography-mass spectrometry (HPLC-MS/MS) proteomic and bioinformatics analyses were performed to explore the molecular mechanisms of ropivacaine treatment in lung cancer cells and to identify new therapeutic targets for ropivacaine. Our study provides evidence that ropivacaine inhibits the malignant behavior of A549 and H1299 lung cancer cells by regulating retinoblastoma binding protein 4 (RBBP4).

Materials & Methods

Cell culture

Human lung cancer cell lines A549 and H1299 were purchased from the Shanghai Institute of Cell Biology (Shanghai, China). Cells were cultured in RPMI-1640 medium (GIBCO, USA) supplemented with 10% bovine serum (Biological Industries, Beit HaEmek, Israel). The monolayer cells were cultivated under controlled conditions of 37 °C temperature and humid air with 5% CO2 supplementation. Ropivacaine was obtained from AstraZeneca AB (Sweden), dissolved in saline with pH adjusted to 7.4, and stored at −20 °C. The ropivacaine concentration and duration was selected as described in our previous study (Shen et al., 2022).

Protein extraction and digestion

After incubating with 1 mM ropivacaine or saline for 48 h, proteins of A549 cells were extracted, then sonicated for six cycles of 5s on, 5s off with RIPA buffer added, and denatured at 95 °C for 2 min. To prepare for proteomic experiments, we removed insoluble fragments were removed from the supernatant by centrifugation at 12,000 × g for 10 min. Protein concentrations were measured using a BCA kit (Thermo Fisher Scientific, Waltham, MA, USA).

Protein digestion was performed following the filtration-assisted sample preparation (FASP) procedure. Disulfide bonds were disrupted through the use of 50 mM DTT in 300 µL UA buffer (0.1 M HCl containing 8 M urea, pH 8.5) for a duration of 30 mins at 37 °C. After digestion, peptides were extracted by centrifugation, lyophilization, and acidification with 0.1% FA. The peptide concentrations were determined using the BCA peptide quantification kit (Thermo Fisher Scientific).

Proteome analysis

To detect protein changes and determine the underlying mechanism of the antitumor effect of ropivacaine, we performed liquid chromatography–mass spectrometry/mass spectrometry analysis (HPLC-MS/MS), as previously described (Wei et al., 2022). Briefly, 1 µg peptides from each sample were loaded onto the nanoflow HPLC EasynLC1200 system (Thermo Fisher Scientific) for proteomic analysis, using a 90-min LC gradient at 300 nL/min. The buffer systems consisted of 0.1%(v/v) FA in H2O for Buffer A and 0.1% (v/v) FA in 80% acetonitrile for Buffer B. The gradient parameters were established as 2–8% B for 1 min, 8–28% B for 60 min, 28–37% B for 14 min, 37–100% B for 5 min, and 100% B for 10 min. The Q exact HF mass spectrometer (Thermo Fisher Scientific) was utilized for this analysis, with a spray voltage of 2100 V and an ion transfer tube temperature of 320 °C set in positive ion mode. Data acquisition was performed using Xcalibur software, with a particular focus on the profile spectrum data type.

The resolution of the full MS1 scan was set at 60,000 m/z 200, AGC target 3e6, maximum IT 20 ms, using the Orbital Trap Mass Analyzer (350–1,500 m/z), followed by the “first 20” MS2 scans from higher energy collisional dissociation (HCD) debris with a resolution of 15,000 at m/z 200, AGC target 1e5, maximum IT 45 ms. The MS2 spectrum’s fixed first mass was set to 110.0 m/z, while the isolation window was set to 1.6 m/z. The normalized collision energy (NCE) was set to 27%, and the dynamic exclusion time was 45 s. Precursors with charges of 1, 8, and >8 were excluded from the MS2 analysis.

Identification of differentially expression proteins (DEPs) and hub genes

The initial processing of data was executed in Proteome Discoverer 2.2, utilizing label-free quantification that relied on ion currents or a methodology akin to the fundamental protein identification technique previously delineated (Shen et al., 2017). Subsequently, we carried out differential protein expression analysis between the ropivacaine-treated and control groups to identify DEPs. Significance was then determined by analysis of variance based on the peptide background at both the peptide group and protein levels (Oberg & Vitek, 2009). The cutoff criteria for DEPs were —log2(Fold Change)—>0.58 & p-value <0.05

The DEPs were subjected to screening in the STRING online database (https://string-db.org/) to establish a protein–protein interaction (PPI) network and ascertain comprehensive interactions. Subsequently, a differential gene interaction network map was generated. The resulting interactive network data was then exported to Cytoscape 3.9.1 software for the identification of the network’s central node protein using various algorithms.

Bioinformatics analysis of RBBP4 from public database

After screening ropivacaine-induced DEPs in A549 cells using proteomic analysis, RBBP4, a downregulated DEP, was selected for further exploration. We downloaded mRNA expression data for the cancer genome atlas (TCGA) lung adenocarcinoma samples (TCGA-LUND) and GTEx healthy lung tissues from the Xiantaoxueshu database (https://www.xiantaozi.com/) (Tang et al., 2017). All data profiles were normalized using log2(x+1) transformation. The R “limma” package was used to perform differential expression analysis between the normal and tumor tissues. Volcanoes and heat maps were generated using the ggplot2 package in R. Furthermore, the protein levels of RBBP4 in lung adenocarcinoma and normal lung tissues were detected using immunohistochemistry images from the Public Database of The Human Protein Atlas (https://www.proteinatlas.org/). We then conducted a survival analysis to assess the correlation between the mRNA expression of RBBP4 and survival profiles using the Kaplan–Meier plotter platform (http://www.kmplot.com) (Győrffy et al., 2013). Receiver operating characteristic (ROC) curves were generated to identify the diagnostic significance of RBBP4 in lung cancer using the XIANTAO Database.

Western blot analysis

Proteins were extracted from cells subjected to various treatments utilizing RIPA buffer supplemented with protease inhibitors. The resulting lysates were centrifuged, and the proteins were denatured by heating. Protein concentrations were determined using a BCA assay (Beyotime, Nanjing, China). Subsequently, 40 µg of total protein was separated using 10% SDS-PAGE and subsequently transferred onto polyvinylidene difluoride (PVDF) membranes (Millipore, MA, USA). The membranes were blocked with 5% BSA serum albumin for 2 h at room temperature, followed by overnight incubation at 4 °C with antibodies specific to RBBP4 or GAPDH. Anti-rabbit IgG (Cell Signaling Technology, Boston, MA, USA) was detected using Odyssey imaging (LI-COR, Lincoln, NE, USA). Antibodies targeting RBBP4 were procured from Beyotime Biotechnology Co., Ltd. (Nanjing, China), whereas antibodies targeting GAPDH were obtained from Abcam (Cambridge, UK).

RNA isolation and quantitative real-time PCR

Briefly, total RNA was extracted using Ultrapure RNA Kit (Cwbio). cDNA was obtained through reverse transcription using TaKaRa PrimeScript RT reagent Kit (TaKaRa). The expression status of RBBP4 and GAPDH was determined by quantitative real-time PCR, using the QuantStudio 6 Flex Real-Time PCR system (Thermo Fisher Scientific). We quantified the relative expression of each target gene for three times using the ΔΔCt method. Primers used were as follows: RBBP4 forward primer: 5′-ATGACCCATGCTCTGGAGTG-3′, and RBBP4 reverse primer: 5′-GGACAAGTCGATGAATGCTGAAA-3′. GAPDH forward primer: 5′-ATG GGG AAG GTG AAG GTC G-3′, GAPDH reverse primer: 5′-GGG GTC ATT GAT GGC AAC AAT A-3′.

Plasmids, SiRNA, and transfection

Plasmids encoding the human RBBP4 gene and two siRNA oligonucleotides targeting human RBBP4 (si-RBBP4-1 and si-RBBP4-2), corresponding to the si-control, were procured from Ruiying Biotech (Changsha, China). The constructs and controls were transfected using Lipofectamine 2000 (Invitrogen, Carlsbad, CA, USA) according to the manufacturer’s instructions and previous literature (Li, Lv & Zhu, 2020). In brief, 2 × 105–3 × 105 cells were transfected with 100 pmol siRNA or 2 µg plasmid DNA. Western blotting was used to detect the transfection efficiency 48 h after transfection, and real-time polymerase chain reaction was used for verification. The siRNA sequences are listed in Table 1.

Table 1 The sequences of siRNA for RBBP4.

Number	Gene	Sequences	
1	Si-1	CCUUCUAAACCAGAUCCUUTT
AAGGAUCUGGUUUAGAAGGTT	
2	Si-2	GCUCAAGUGAACUGCCUUUTT
AAAGGCAGUUCACUUCAGCTT	

Cell proliferation assays

A total of 5 × 103 cells/well were seeded into 96-well plates and subjected to treatment with 1 mM ropivacaine and 100 µL DMEM supplemented with 10% FBS for 24, 48, 72, and 96 h. At specific time intervals, 20 µL of MTS [3-(4,5-dimethylthiazol-2-yl)-5-(3-carboxymethoxyphenyl)-2-(4-sulfophenyl)-2H-tetrazole] solution (Promega, Madison, WI, United States) was added to the cells and incubated for 2 h at 37 °C. Subsequently, the cells were subjected to enzymatic labeling (Thermo Fisher Scientific, Waltham, MA, United States) to determine the absorbance at 492 nm.

Transwell assay

The study employed Boyden chambers (pore size: 8 µm) obtained from Collaborative Biomedical, Becton Dickinson Labware, Bedford, MA, USA, either covered or uncovered with 200 µg/mL matrigel (Beyotime Biotechnology), to evaluate the migration and invasion capabilities of A549 or H1299 cells ( 1 ×105). The upper chamber was inoculated with cells in 0.2 mL of RPMI 1640 medium without serum, while the lower chamber contained 0.6 mL of medium with 10% FBS. Following an 18-h incubation period, non-migrating cells were removed from the membrane using a cotton swab, and crystal violet was applied to the submembrane-permeating cells. The number of cells that penetrated the membrane was determined by microscopic observation of five randomly selected regions.

Cell-cycle analysis

We used flow cytometry and propidium iodide (PI) staining to evaluate the effect of ropivacaine on cell cycle progression in A549 and H1299 cells. Following inoculation in six-well plates, the cells were incubated for 24 h and subsequently exposed to serum-free medium for an additional 24 h to synchronize the G0/G1 phase of the cell cycle. The cells were then treated with ropivacaine (1 mM) for 48 or 72 h. Subsequently, the cells were stained with PI (MULTI SCIENCES, Hangzhou, China) according to the manufacturer’s guidelines. The stained cells were then subjected to incubation at 37 °C for 20 min, followed by analysis using a FACSCalibur flow cytometer (BD Biosciences, Franklin Lake, NJ, USA). Approximately 5 × 105 cells were collected, followed by two washes with phosphate-buffered saline (PBS) and subsequent resuspension with 500 µL 1 × binding buffer. The solution was supplemented with FITC membrane-linked protein V and PI iodide and incubated for 5 min at room temperature in the dark. Flow cytometric analysis was performed on the cells, which were gently vortexed prior to analysis.

Statistical analysis

Statistical analyses were conducted and visualized using SPSS version 26.0 (IBM Corp, Armonk, NY, USA) and GraphPad Prism version 9.0 (GraphPad Software Inc., San Diego, CA, USA). Data are presented as the mean ± standard deviation of three independent experiments. To enable comparisons between multiple groups, one-way analysis of variance (ANOVA) or the Kruskal–Wallis H test was performed, followed by Tukey’s post hoc tests. Differences between RBBP4 overexpression groups were calculated using two-way ANOVA. Chi-square or Fisher’s exact test was used for categorical variables to determine differences between the groups. A p-value <0.05 was considered statistically significant.

Results

DEPs identification and bioinformatics analysis

Lung cancer A549 cells were co-cultured with ropivacaine (1 mM) or normal saline and subjected to HPLC-MS/MS proteomic analyses to explore the mechanism and potential therapeutic targets of ropivacaine in lung cancer (Fig. 1A). Figure 1B shows the results of the principal component analysis. A total of 2,672 proteins were quantified following ropivacaine treatment. Of these, 142 were significantly downregulated, while 185 were significantly upregulated DEPs with the cutoff —log2(Fold Change)—>0.58 & p-value <0.05 (Table S1). The data are presented as a volcano plot (Fig. 1C) and a clustered heatmap (Fig. 1D).

Figure 1 Quantitative proteomic analysis of control and ropivacaine-treated A549 cells and bioinformatics analysis of differentially expressed proteins.

(A) The flow chart of the HPLC-MS/MS analysis to screen DEPs and identify hub genes. (B) Principal component analysis of the six samples from control and ropivacaine-treated groups. (C) Volcano plots of the DEPs between two groups, with red dots indicating high expression and green dots indicating low expression. (D) Heatmap of the DEPs between normal saline and ropivacaine-treated groups, with red indicating high expression, blue indicating low expression, NC representing normal saline group, and sample representing ropivacaine-treated groups. (E) The PPI network of DEPs was constructed using STRING and Cytoscape. (F) Five algorithms (MCC, MNC, Degree, Closeness and BottleNeck) were employed to search for top 10 hub genes. (G) these hub genes were analyzed by Venn diagram, and the candidate mitochondrial gene (RBBP4) was finally confirmed. Abbreviations: DEPs, differentially expressed proteins; HPLC-MS/MS, high-performance liquid chromatography-mass spectrometry; PPI, Protein–protein interaction.

The PPI network of the DEPs was constructed using STRING and Cytoscape (Fig. 1E). Five algorithms (MCC, MNC, Degree, Closeness and BottleNeck) were employed to search for the top 10 hub genes (Fig. 1F). The top 10 hub genes identified by the MCC algorithm were RBBP4, ETFA, RPS29, ACAA2 S25, ACAT2, ACOX1, HADHB, ACAT1, HADHA. The top 10 hub genes identified by the MNC algorithm were RBBP4, NDUFA4, TCERG1, RPS25, MDH2, RPS29, NDUFA6, RPL36A, SRRM1, UQCRC2. The top 10 hub genes identified by the Degree algorithm were RBBP4, CYCS, UQCRC2, ACAT2, SRRM1, RPS25, MDH2, RPS29, NCBP1, ACOX1. The top 10 hub genes identified by the Closeness algorithm were RBBP4, NDUFA4, CYCS, CYB5A, UQCRC1, NDUFS7, RPS29, MDH2, ETFA, UQCRC2. The top 10 hub genes identified by the BottleNeck algorithm were RBBP4, PTGS2, SLC25A6, UBE2I, CYCS, TRIM28, SLC25A3, SRC, UQCRC2, CBX1.These hub genes were analyzed using a Venn diagram (Fig. 1G). Finally, RBBP4, a downregulated DEP, was selected for post-validation among the candidates and is considered to play a crucial role in the antitumor effect of ropivacaine in lung cancer cells.

RBBP4 expression is upregulated in LUAD tissues and its high expression is correlated with poor prognosis

To fully investigate the potential involvement of RBBP4 in the pathogenesis of lung cancer, we first analyzed the expression levels of this gene across 33 cancer datasets obtained from the TCGA database. As shown in Fig. 2A, the expression of RBBP4 was significantly increased in 13 of the 33 cancer types. Additionally, RBBP4 expression was significantly elevated in TCGA-LUAD tissues compared to controls (p < 0.001, Figs. 2B–2E). These results suggested that elevated RBBP4 expression may induce LUAD progression.

Figure 2 RBBP4 was overexpressed in human lung cancer tissues and was associated with poor prognosis.

(A) RBBP4 expression was significantly upregulated in many cancers, including lung adenocarcinoma (LUAD). The comparison of RBBP4 expression levels in 33 cancerous tissues and their corresponding adjacent normal tissues from the TCGA database. (B) The TCGA database showed that RBBP4 expression was significantly elevated in LUAD tissues compared to normal tissues. (C) The comparison of RBBP4 expression difference between paired samples in TCGA LUAD cohort. The protein level of RBBP4 in lung normal tissues (D) and lung adenocarcinoma tissues (F). (F-H) The overall survival (OS), first progression (FP), and post-progression survival (PPS) rates in RBBP4-high and RBBP4-low patient groups from Kaplan-Meier Plotter database. Red and black curves represent LUAD patients with high and low RBBP4 expression, respectively. (I) Diagnostic ROC curves differentiating LUAD tissue from normal tissue based on RBBP4 expression levels. Abbreviations: ACC, adrenocortical carcinoma; BLCA ,bladder urothelial carcinoma; BRCA, breast invasive carcinoma; CESC, cervical squamous cell carcinoma and endocervical adenocarcinoma; CHOL, cholangiocarcinoma; COAD, colon adenocarcinoma; DLBC, lymphoid neoplasm diffuse large B-cell lymphoma; ESCA, esophageal carcinoma; GBM, glioblastoma multiforme; HNSC, head and neck squamous cell carcinoma; KICH, kidney chromophobe; KIRC, kidney renal clear cell carcinoma; KIRP, kidney renal papillary cell carcinoma; LAML, acute myeloid leukemia; LGG, brain lower grade glioma; LIHC, liver hepatocellular carcinoma; LUAD, lung adenocarcinoma; LUSC, lung squamous cell carcinoma; MESO, mesothelioma; OV, ovarian serous cystadenocarcinoma; PAAD, pancreatic adenocarcinoma; PCPG, pheochromocytoma and paraganglioma; PRAD, prostate adenocarcinoma; READ, rectum adenocarcinoma; SARC, sarcoma; SKCM, skin cutaneous melanoma; STAD, stomach adenocarcinoma; TGCT, testicular germ cell tumors; THCA, thyroid carcinoma; THYM, thymoma; UCEC, uterine corpus endometrial carcinoma; UCS, uterine carcinosarcoma; UVM, uveal melanoma. FP, first-order progression; OS, overall survival; PPS, post-progression survival. ∗p < 0.05, ∗∗p < 0.01, ∗∗∗p < 0.001.

To determine the potential prognostic significance of RBBP4 in lung cancer, we examined the association between RBBP4 expression and overall survival (OS), first progression survival (FP), and post-progression survival (PPS) using the Kaplan–Meier mapper plotter database. Our results of Kaplan–Meier survival curve analysis showed that patients with LUAD with high-RBBP4 expression had significantly lower OS (HR =1.54, p < 0.001, Fig. 2F), FP (HR =1.82, p < 0.001, Fig. 2G), and PPS (HR =1.48, p = 0.018, Fig. 2H) than those with low-RBBP4 expression levels. Finally, we evaluated the diagnostic significance of RBBP4 in LUAD by plotting ROC curves. Our data show that RBBP4 had good predictive accuracy for diagnosing LUAD, with an area under the curve (AUC) of 0.743 (95% CI [0.702–0.785], Fig. 2I). These results showed that higher expression of RBBP4 may be related to the poor prognosis of patients with lung cancer.

RBBP4 knockdown receded proliferation, migration, invasion, blocked cell cycle of lung cancer cells

To examine the biological role of RBBP4 in lung cancer cells, we used siRNA-mediated gene silencing to knock down RBBP4. Subsequently, western blotting and RT-qPCR were performed to evaluate the expression of RBBP4 in A549 and H1299 cells transfected with either Si-NC, Si-RBBP4-1 and Si-RBBP4-2. The protein expression of RBBP4 was downregulated in both the Si-RBBP4-1 and Si-RBBP4-2 groups compared to that in the Si-NC group in A549 and H1299 cells (Fig. S1).

The MTS assay results showed that RBBP4 knockdown by siRNAs led to a significant decrease in the proliferation of A549 and H1299 cells (Figs. 3A–3B). We further investigated the impact of RBBP4 knockdown on lung cancer cell migration and invasion using Transwell assays. The results showed a reduction in the number of migrated and invaded cells in RBBP4 knockdown groups compared to that in the Si-NC group (Figs. 3C and 3D). Considering that RBBP4 knockdown suppressed the proliferation of A549 and H1299 lung cancer cells, we further explored its effect on the cell cycle. Our flow cytometry results revealed a significant increase in the proportion of cells in the G0/G1 phase in the Si-RBBP4-1 and Si-RBBP4-2 groups compared to that in the Si-NC group (Fig. 3E).

Figure 3 RBBP4 knockdown inhibits tumor-like behavior in lung cancer cells.

(A, B) The impact of RBBP4 knockdown on the proliferation of A549 and H1299 cells was assessed through an MTS assay. (C–D) The transwell migration/invasion assay was used to analyze the effect of RBBP4 knockdown on cell migration and invasion ability (scar bar = 200 µm). (E) The impact of RBBP4 knockdown on cell cycle distribution was measured using PI staining and flow cytometry. Data were presented as the mean ±standard deviation of three independent experiments. ∗p < 0.05; ∗∗p < 0.01; ∗∗∗p < 0.001 vs. the Si-NC group.

These findings demonstrate the crucial role of RBBP4 in the malignant behavior of lung cancer cells. In the present study, we successfully suppressed RBBP4 expression in A549 and H1299 cells using siRNAs.

RBBP4 knockdown enhances the antitumor effect of ropivacaine

Our proteomic results and PPI analysis results showed that RBBP4, one of downregulated DEP, was the hub gene. Western blotting was used to verify RBBP4 expression in A549 cells treated with ropivacaine (0, 0.5, 1, and 2 mM). Our results showed that RBBP4 protein expression levels were significantly downregulated in a contraction-related manner in the ropivacaine-treated group compared to the control group (Fig. 4A), indicating that ropivacaine reduced RBBP4 expression in lung cancer cells.

Figure 4 RBBP4 knockdown enhances the antitumor effect of ropivacaine (Rop).

(A) The lung cancer A549 cells were subjected to incubation with different concentrations of ropivacaine (0, 0.5, 1, or 2 mM), and the alterations in RBBP4 expression were determined through western blot analysis. (B) The lung cancer A549 and H1299 cells were subjected to transfections with either Si-NC or Si-RBBP4, followed by incubation with or without 1 mM ropivacaine. Subsequently, the proliferation of cells was assessed through the MTS assay. (C–D) The effect of ropivacaine combined with RBBP4 knockdown on cell migration and invasiveness was measured using a transwell assay. The number of cells that migrated or invaded was counted in five different fields. Representative images (scar bar = 200 µm) to the left, quantification of migrated or invaded cells to the right. (E) The changes in cell cycle distribution measured using Flow cytometry. Data are presented as the mean ±standard deviation of three independent experiments. ∗p < 0.05, ∗∗p < 0.01, ∗∗∗p < 0.001 vs the control group.

After confirming the silencing effects of Si-RBBP4-1 and Si-RBBP4-2, we investigated the effect of ropivacaine combined with RBBP4 knockdown on the malignant behavior of A549 and H1299 cells using the MTS, Transwell, and flow cytometry assays. Our results indicated that ropivacaine inhibited the proliferation, migration, and invasion of A549 and H1299 cells. Interestingly, compared with ropivacaine (Rop) group, the proliferation, migration, and invasion capabilities of the Rop+Si-RBBP4-1 and Rop+Si-RBBP4-2 group were also significantly downregulated (Figs. 4B–4D). Consistent with the changes in MTS, the results from flow cytometry showed that RBBP4 knockdown combined with ropivacaine treatment induced cell cycle arrest at the G0/G1 phase in lung cancer cells, which was the most remarkable among the four groups (Fig. 4E). These findings demonstrated that RBBP4 knockdown enhanced the antitumor effects of ropivacaine.

RBBP4 overexpression attenuated the inhibitory effect of ropivacaine on A549 and H1299 cells

To explore the effect of RBBP4 overexpression on the antitumor effect of ropivacaine in lung cancer cells, A549 and H1299 cells were transfected with RBBP4 overexpression plasmids. Western blotting and RT-qPCR were performed to evaluate the expression of RBBP4 in A549 and H1299 cells transfected with either NC or RBBP4 overexpression. The protein expression of RBBP4 was upregulated in RBBP4 overexpression groups compared to that in the NC group in A549 and H1299 cells (Fig. S2). MTS, transwell, and flow cytometry analyses showed that RBBP4 overexpression enhanced the proliferation, migration, and invasion of A549 and H1299 cells and reduced the proportion of cells in G0/G1. The inhibitory effect of ropivacaine on A549 and H1299 cells was significantly reversed in the Rop+RBBP4 overexpression group compared to the Rop group (Figs. 5A–5C). Furthermore, flow cytometric analysis revealed that the proportion of G0/G1 phase cells was higher in the ropivacaine group than in the control group, whereas the proportion of G0/G1 phase cells was lower in the Rop+RBBP4 overexpression group than in the Rop group (Fig. 5D). These findings suggest that ropivacaine inhibits the malignancy of lung cancer cells by suppressing RBBP4 expression.

Figure 5 Overexpression of RBBP4 promotes the proliferation, migration and invasive ability of lung cancer cells and eliminates the antitumor effect of ropivacaine.

A549 and H1299 cells were co-transfected with negative control, ropivacaine, and pcDNA-RBBP4 with or without 1 mM ropivacaine. (A) The MTS assay was utilized to quantify cell proliferation. (B–C) The Transwell assay was utilized to detect cell migration and invasion ability (scar bar = 200 µm). (D) The changes in the cell cycle distribution were assessed using flow cytometry. Data are presented as the mean ±standard deviation of three independent experiments. ∗p < 0.05, ∗∗p <0.01, ∗∗∗p < 0.001 vs the control group.

Discussion

The present study provides evidence that ropivacaine treatment inhibits lung cancer cell proliferation, migration, and invasion, and induces cell cycle arrest. RBBP4 is one of the downregulated DEPs and confirmed as hub gene for further validation. High RBBP4 expression predicts poor survival outcome in patients with lung cancer. Overexpression and knockdown of RBBP4 significantly altered the antitumor effects of ropivacaine on the malignant behavior of lung cancer cells.

Mass spectrometry-based proteomic technologies have become important tools for identifying cellular targets of drug action. We used HPLC-MS/MS proteomics and bioinformatic approaches to identify DEPs and the potential mechanism of action of ropivacaine in A549 cells. Then, we performed PPI analysis to find hub genes, and RBBP4 was finally confirmed to be the candidate hub gene.

RBBP4 is a recently discovered protein with tumor-specific characteristics and a molecular weight of 48 kDa. Its nomenclature is derived from its ability to interact with retinoblastoma proteins both in vivo and in vitro (Tsujii et al., 2015). RBBP4 plays a crucial role in various types of cancers. Dysregulated expression of RBBP4 is associated with metastasis and unfavorable prognosis in highly aggressive and metastatic cancers, such as non-small cell lung cancer cells (Wang et al., 2021; Cao et al., 2021). Tumorigenesis is also suppressed following the inhibition of RBBP4 expression. Additionally, RBBP4 overexpression enhances tumor cell radiosensitivity by inhibiting the PI3K/Akt pathway (Jin et al., 2018). Analysis of the TCGA database revealed that heightened expression of RBBP4 was associated with unfavorable survival outcomes in patients with lung cancer. Concurrently, in vitro cytological investigations revealed that suppression of RBBP4 led to a reduction in the proliferation and invasiveness of lung cancer cells. Conversely, the upregulation of RBBP4 demonstrated the opposite effect, indicating its oncogenic role in the progression of lung cancer. Consistent with our investigation, the suppression of RBBP4 impeded the proliferation, invasion, and migration of triple-negative breast cancer cells by modulating the epithelial-mesenchymal transition (Zheng, Yao & Liu, 2022), while the promotion of colon cancer progression was facilitated by RBBP4 through the augmentation of the Wnt/ β catenin pathway (Li, Lv & Zhu, 2020). These findings suggest that oncogenic RBBP4 has potential as a therapeutic target for cancer treatment.

Figure 6 The schematic diagram depicted the antitumor effect and potential mechanism of ropivacaine on lung cancer cells in vitro.

Retinoblastoma binding protein 4 (RBBP4) was overexpressed in lung cancer tissues and predicted poor survival outcomes for lung cancer patients. RBBP4 knockdown significantly suppressed the cell proliferation, migration and invasion capacity and induced cell cycle arrest of lung cancer A549 and H1299 in vitro. Ropivacaine inhibited the malignant behavior of lung cancer cells by regulating the expression of RBBP4. The schematic diagram was designed using FigDraw (http://www.figdraw.com).

Prior research has demonstrated that ropivacaine affects the biological behavior of lung adenocarcinoma cells; however, these findings are controversial and require further investigation (Piegeler et al., 2015; Wang et al., 2016). Ropivacaine treatment was shown to have beneficial antimetastatic effects on NCI-H838 lung cancer cells, potentially by suppressing tumor necrosis factor-α-induced src-activation and intercellular adhesion molecule-1 phosphorylation (Piegeler et al., 2012). A previous study also reported that the cytotoxic effect of ropivacaine on human non-small cell lung cancer involves the apoptotic and MAPK pathways (Wang et al., 2016). These studies highlight the importance of cell signaling pathways and molecules as potential new targets for the antitumor activity of ropivacaine. The present study aimed to determine whether ropivacaine modulates the biological function of lung cancer cells through RBBP4. Western blot analysis demonstrated that ropivacaine treatment led to a contraction-related reduction in RBBP4 expression in A549 cells, which is consistent with the proteomics data. Moreover, RBBP4 knockdown by siRNA enhanced the proliferation, migration, and invasive capacity of H1299 and A549 cells in response to ropivacaine, while impeding the cell cycle. In contrast, overexpression of RBBP4 using plasmids attenuated these inhibitory effects. These ropivacaine-induced changes may be associated with reduced RBBP4 expression, thus providing a novel theoretical reference and scientific basis for the use of ropivacaine in the treatment of lung cancer (Fig. 6).

Our study had several limitations. First, the appropriate integration of proteomic data with other omics types (genomics, transcriptomics, and metabolomics) can help elucidate the complex molecular mechanisms of cancers and facilitate the development of novel drugs (Xu et al., 2020). Our team will conduct multi-omics analyses to further explore the antitumor effects of ropivacaine. Second, a normal lung cell line was used as a control to detect ropivacaine-induced normal tissue toxicity. Ropivacaine has been shown to suppress the wound healing rate and keratinocyte proliferation and migration in a rat model (Wu et al., 2022). Thirdly, the present study could not demonstrate the targeting relationship between ropivacaine and RBBP4 protein, which need more experiments to validate, such as molecular docking technology. Finally, our work was an in vitro cytological experiment that does not accurately represent clinical conditions. Therefore, further clinical trials are required to determine the therapeutic effects of ropivacaine in patients with lung cancer undergoing surgical resection.

Conclusions

Ropivacaine inhibited the proliferation, invasion, and migration of A549 and H1299 cells and blocked the cell cycle by regulating the expression of RBBP4. Our study provides a better understanding of the tumor-suppressive effects of ropivacaine, which need to be validated by further studies.

Supplemental Information

Supplemental Information 1 DEPs between ropivacaine-treated groups and control groups in A549 cells

Click here for additional data file.

Supplemental Information 2 Raw data for the figures

Click here for additional data file.

Supplemental Information 3 Raw data for supplementary figures

Click here for additional data file.

Supplemental Information 4 Validation of RBBP4 expression after knockdown lung cancer A549 (A) and H1299 (B) lines using RT-qPCR and Western blot

Click here for additional data file.

Supplemental Information 5 Validation of RBBP4 expression after overexpression in lung cancer A549 (A) and H1299 (B) lines using RT-qPCR and Western blot

Click here for additional data file.

The authors thank the Fourth Hospital of Hebei Medical University for their assistance with the manuscript submission.

Additional Information and Declarations

Competing Interests

Author Contributions

Data Availability

The authors declare there are no competing interests.

Weiai Jia performed the experiments, analyzed the data, prepared figures and/or tables, and approved the final draft.

Junmei Shen performed the experiments, prepared figures and/or tables, and approved the final draft.

Sisi Wei performed the experiments, prepared figures and/or tables, and approved the final draft.

Chao Li analyzed the data, prepared figures and/or tables, and approved the final draft.

Jingpu Shi analyzed the data, prepared figures and/or tables, and approved the final draft.

Lianmei Zhao analyzed the data, authored or reviewed drafts of the article, and approved the final draft.

Huiqun Jia conceived and designed the experiments, authored or reviewed drafts of the article, and approved the final draft.

The following information was supplied regarding data availability:

The raw data is available in the Supplemental Files.

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
