# Peer review of "Ropivacaine inhibits the malignant behavior of lung cancer cells by regulating retinoblastoma-binding protein 4"

_PeerJ, doi:10.7717/peerj.16471_

## Round 0.1 · original submission · Major Revisions

Dear Dr. Jia,

Thank you for submitting your manuscript "Ropivacaine inhibits the malignant behavior of lung cancer cells by regulating RBBP4" to PeerJ. We have now received reports from the reviewers, and, after careful consideration, we have decided to invite a major revision of the manuscript.

As you will see from the reports copied below, the reviewers raise important concerns. We find that these concerns limit the strength of the study, and therefore we ask you to address them with additional work. Without substantial revisions, we will be unlikely to send the paper back for review.

If you feel that you are able to comprehensively address the reviewers’ concerns, please provide a point-by-point response to these comments along with your revision. Please show all changes in the manuscript text file with track changes or color highlighting. If you are unable to address specific reviewer requests or find any points invalid, please explain why in the point-by-point response.

Thanks

Abhishek Tyagi, PhD
Academic Editor,
PeerJ

Reviewer 1 ·

Basic reporting

The manuscript is overall clear! The background is well described in the introduction to provide sufficient context. However, some concerns in basic reporting may need to be addressed. Specific comments are provided below:
- Figures 1B, 1C, 1E, 1F, 3E, 4E, 5D all have illegible text labels. This significantly hinders the interpretation of the figures. Please check and revise as necessary using larger texts.
- RBBP4 was not well introduced. It was mentioned in lines 239-240: "150 were significantly down-regulated while 199 were significantly up-regulated DEPs". It is suggested to provide additional support for why RBBP4 was selected out of hundreds of possible candidates for investigation. Also, for Lines 256-257: "taking into account the fold change and relevant literature" – it was not clear what literature this is referring to – please elaborate.
- The terms "gene" and "protein" are sometimes used interchangeably in this manuscript. It would be helpful to exercise more caution when discussing the differences between gene and protein readouts in the experiments and analysis.
- Line 260: "17 out of 33 cancerous tissue samples" – this does not seem right. Figure 2A appears to be showing different types of cancers, and within each type, there are numerous samples – hence the description that they are "33 cancerous tissue samples" is incorrect. Also, I counted 13 marked significant, not 17. Please check and clarify this section as necessary. It is also recommended to have a legend for the abbreviations shown in Figure 2A.
- For Figure 2F-H, the numbers at risk were not legible.
- For Figures 3A and 4A, please indicate what the error bars in the graph on the right represent.
- The images presented for H1299 in Figures 3C and 3D appear identical, yet they are labeled as representing "Migration" and "Invasion" respectively. While the observed trends in these figures are the same, the reported cell counts differ. Please check and verify that the correct image was placed.
- It is recommended to provide more information on the siRNA in the figure or figure captions, including which gene is being knocked down, how long post-transfection the assays were conducted, etc.

Experimental design

- The statistical analysis performed on the DEPs is unclear. Line 152: "cutoff criteria for DEPs are |log2(Fold Change)| > 0.58 & p value < 0.1" – the selection of p-value < 0.1 may not be stringent enough due to the high number of comparisons made. It may help to provide some justifications for the selection of the cutoffs. Furthermore, the methodology for calculating the p-value was not detailed in the manuscript – it remains ambiguous whether the reported p-value has been adjusted for multiple comparisons.
- It is unclear whether a 48-hour period is a sufficient time for Ropivacaine treatment to result in protein level changes. It is recommended to justify the selection of the 48h time point, either from the literature or experiments.
- Line 192-193: "while validation was performed through Transwell and cell cycle analyses" – please provide more information on what validation was carried out.
- The efficacy of some siRNA knockdowns was not significant. Optimizing the transfection or testing alternative target sequences is recommended to ensure that RBBP4 is significantly knocked down, particularly in "Si-2", where the efficiency in both cell lines is particularly low.
- Line 190 describes a 48-hour transfection, but there seems to be no clear justification for this timeframe. It's crucial to optimize this process as knockdowns usually take time to be reflected at the protein level, depending on various factors, such as protein half-life. It would be helpful to provide further justification as to if 48 hours is sufficient, considering the insufficient knockdown efficiency as noted above.
- For all experiments discussed in the Methods, please indicate the number of biological replicates or independent experiments performed. This is particularly missing from the proteome analysis.

Validity of the findings

There are some serious concerns about the validity of some findings. Please see specific comments below:
- Line 245-248: "The KEGG pathway analysis revealed that oxidative phosphorylation, RNA transport, PI3K-AKT signaling pathway, HIF-1 signaling pathway, and VEGF signaling pathway were significantly altered" – these pathways do not seem to have direct relevance to RBBP4. Also, although the pathways noted are on the list of significant pathways in Figure 1E, they are not the most significant or have the highest counts. This appears to be a selective representation and may be misleading. Please check and adjust as necessary.
- The chosen significance cutoff (p-value < 0.1) for DEPs seems to lack stringency. Taken together with the point above, it is recommended to avoid excessive interpretations from the enriched pathways identified via the DEPs (Line 243-252, 340-349), especially considering there aren't any follow-up validation experiments performed to confirm the role of Ropivacaine.
- Si-2 siRNA’s knockdown efficiency in A549 cells was not clear, as shown in Figure 3A. Yet, in Figure 3B, the effect on proliferation is roughly in line with Si-1, which has reduced RBBP4. This is conflicting, and the experiment may need to be further validated. Please refer to points on siRNA knockdown efficiency in the comments for "Experimental Design."
- As previously discussed in Experimental Design, some siRNA knockdowns are not efficient. This may call into question the validity of all experiments that use these siRNA knockdowns in Figure 3-4. Please demonstrate the validity of the knockdowns or use an alternative target sequence to ensure that the knockdowns are effective in the experiments conducted.
- To validate the overexpression system, it is necessary to have a western blot showing the increase in RBBP4.
- Line 219-321: “H1299 cells were significantly reversed in the RBBP4 overexpression + ropivacaine group when compared to the ropivacaine group (Figure 5A-C)”. In Figure 5A: RBBP4 overexpression vs. RBBP4 overexpression + Ropivacaine do not seem very different. Please validate the finding again and provide data that clearly demonstrates this.
- In Figure 4B – the Si-NC + Ropivacaine and Si-1 + Ropivacaine do not appear significantly different in the proliferation plot. This does not support the claim that "the reduction of proliferation, migration, and invasion capabilities in Si-1+ ropivacaine group was more significant than that in the ropivacaine group" (lines 306-308).
- It is recommended to perform all biological assays with two valid siRNA target sequences to ensure reproducibility.

Additional comments

The authors could also consider using String DB <https://string-db.org/> to look at potential interaction networks in the DEPs.

Reviewer 2 ·

Basic reporting

The biggest problem of this manuscript is the poor writing. A lot of grammar problems can be easily found. Here are just some examples in the Abstract and Results sections:

1. Line 40, a “that” missed after “found”
2. Line 41, it should be “explore”, instead of “explored”
3. Line 45, it should be “proteomics was”, rather than “were”
4. Line 51, it should be “related to”, rather than “with”; same with Line 55
5. Line 53, it would be better to say “proteomics analysis identified” than using “found”
6. Line 54, a “be” missed after “may”
7. Line 55, there should be a comma between “transport” and “and”
8. Line 56, there should be a “the” before “downregulated DEPs”, and there should be a “was” after “and”
9. Line 57, it should be “significantly” rather than “significant”
10. Line 58, it should be “western blot results”
11. Line 59, it would be better to use “showed that” or “demonstrated that”, instead of “found”
12. Lines 61 and 62, it is confusing to state that “the antitumor effect of ropivacaine on A549 and H1299 cells was enhanced.” Please rephrase this sentence.
13. Lines 238 and 239, “In comparison to …., 2696 proteins.” This is confusing since this description seems to tell readers that ropivacaine treatment induced 2696 proteins expressed. Please rephrase this sentence.

Experimental design

1. The H1299 images in Figure 3C and 3D seem to be the same. Please double-check the image sources.

2. It would be more valuable if the authors could discuss how Ropivacaine represses RBBP4 protein expression. Does Ropivacaine inhibit Rbbp4 gene expression?

Validity of the findings

This study, conducted by Jia et al., investigates the antitumor effects of ropivacaine, a commonly used local anesthetic in lung cancer surgery, on lung cancer cells. Through quantitative proteomics analysis, the researchers identified 349 differentially expressed proteins (DEPs) triggered by ropivacaine treatment. They highlight the downregulation of retinoblastoma binding protein 4 (RBBP4) as a key finding, supported by its significant upregulation in lung cancer patients and association with poor prognosis. Functional experiments confirm the crucial role of RBBP4 in cell proliferation, invasiveness, and cell cycle regulation. The study reveals that ropivacaine's antitumor effect is enhanced by suppressing RBBP4, while RBBP4 overexpression mitigates ropivacaine's inhibitory impact. This study's findings provide valuable insights into the mechanisms underlying ropivacaine's anticancer properties and offer potential for novel therapeutic strategies, pave the way for improved treatment options, and contribute to the advancement of lung cancer research.

1. The authors used Student t test or One-way ANOVA to compare groups. However, when two levels are tested (e.g. ropivacaine vs vehicle, and siNC vs siRBBP4), a two-way ANOVA should be used, or a non-parametric equivalent (such as Kruskal-Wallis test), if the assumptions of parametric testing are not met. This applies to Figures 4B-4E and Figure 5.

·

Basic reporting

The paper was well-drafted and easy to follow, with only a few minor grammatical errors that need correction.

Experimental design

In general, these experiments were well-designed. But I lack confidence in their data as the author relied heavily on repetitive images and appeared to tally the data haphazardly.

Validity of the findings

no comment

Additional comments

Major Concerns:

1. Regarding Figure 1, there are two issues. Firstly, the figures have a low resolution, making it difficult to see the details clearly. Secondly, the labels on the figures are also hard to read. It would be beneficial to improve the resolution and ensure that the labels are legible.

2. Additionally, I apologize, but I do not have access to the specific rank number of RBBP4 or the reasons for choosing it to study. To provide more information on this, please provide additional context or specific details.

3. In Figure 2, it is important to address the status of other downregulated proteins mentioned in the study. Specifically, it would be helpful to investigate whether these proteins also function as oncogenes. To determine their potential role as oncogenes, further information about the study and the specific proteins in question is necessary. Please provide more details or clarify the context.

4. Figure 3 presents significant concerns. The low knockdown percentage observed raises doubts about the reliability of the subsequent cell proliferation assay and trans well assay. It becomes challenging to place trust in the results when the knockdown efficiency is low. Moreover, reusing images (H1299 cell line) in Figures 3C and 3D is not acceptable.

5. Qualifications were also a major concern. In Figure 3A and 3D upper panel, the images display a significant decrease in cell count, but the qualifications have not been altered. Moreover, the scale bar should be included in these images.

6. For the assessment of cell proliferation, a suggestion has been made to employ Ki67 immunofluorescence (IF) staining.

7. In Figure 4, the question arises: Does ropivacaine impact the mRNA level of RBBP4? Meanwhile, the qualifications do not match with the images presented in Figure 4C and 4D.

8. In Figure 5, a significant concern arises regarding the statistical qualifications for Figure 5C and 5D. The data presented in the images do not correspond appropriately with the statistical results provided. Specifically, in Figure 5D (A549 cell line, upper panel), the number of cells in the Si-1+ ropivacaine group appears to be much higher than in the ropivacaine group. This raises the question of why the qualifications for the Si-1+ ropivacaine group are approximately 2-fold lower than those for the ropivacaine group.

---

## Round 0.2 · Minor Revisions

Dear Dr. Jia,
Thank you for your submission to PeerJ.

The manuscript requires further minor revisions.

Please address these changes and resubmit as required.


With kind regards,
Abhishek Tyagi
Academic Editor
PeerJ Life & Environment

Reviewer 1 ·

Basic reporting

The manuscript has improved significantly, and the authors have addressed most of the comments from the first round of review.
There are a few additional comments on basic reporting that remain:
- In response to Q7 from the previous round of review, Figure 3C-D should have revised images due to a prior mistake by the authors. However, upon closer inspection, H1299 si-NC is again the same image in Figures 3C and D. This is the second occurrence and is very concerning. Although I accept that mistakes happen, especially when handling large sets of images with various file names, it is the author’s responsibility to check the correct image placement and that the analyses of all images were done correctly to ensure the integrity of the research presented. Providing the raw image files in the supplemental files is also suggested.
- Supplemental Figure 1 discusses the various PPI analyses, including determining the “hub genes” and that the “candidate mitochondrial gene (RBBP4) was finally confirmed”. It is recommended to provide slightly more context and explanation around lines 264-269 to describe the analysis and the discovery process more clearly.
- It is recommended to clean up Supplemental Figure 1A. Some suggestions include grouping based on molecular function or pathways and perhaps colored/sized based on the confidence of the hit, etc. It could help provide more insight.
- In response to Q4 from the previous round of review, the Figure 4 caption was revised to include the statement, “Data are presented as the mean ± standard deviation of three independent experiments” – but the Figure 3A caption was not revised.

Experimental design

Additional comments related to experimental design will be discussed alongside comments in the validity of findings section below.

Validity of the findings

The authors addressed most of the concerns raised in the previous round of review. However, there are still some remaining concerns:
- Considering the response to Q2 and that the significant cutoff was set at “|log2(Fold Change)| > 0.58 & p-value < 0.1” for DEP, these cutoffs may not be stringent enough. Although RBBP4 was identified this way and validated by additional experiments, the bioinformatics and pathway analysis (starting line 245) was not. As a result, it is highly recommended to validate the pathways or tone down the findings surrounding the pathway analysis based solely on the DEP data.
- The authors provided qRT-PCR and western blot data showing the knockdowns of RBBP4 (Supplemental Figure 2). For the H1299 samples, siRBBP4-2 was able to achieve a clear knockdown of RBBP4 mRNA in qRT-PCR, but the western blot shows an incomplete knockdown. This appears conflicting. From the phenotype data, the two knockdowns appear to perform similarly.

·

Basic reporting

no comment

Experimental design

no comment

Validity of the findings

no comment

Additional comments

I want to express my gratitude for the authors' dedicated work in completing this manuscript. They have effectively addressed the previous concerns, showcasing their commendable efforts.

---

## Round 0.3 · Minor Revisions

Dear Dr. Jia,

Your submission still requires Minor Revisions.

With kind regards,
Abhishek Tyagi
Academic Editor

Reviewer 1 ·

Basic reporting

Comments from the previous round of review have been addressed. The authors have opted to remove discussions surrounding KEGG/GO analysis due to limitations in resources to perform necessary validation experiments. This is understandable and does not impact the key findings of this manuscript, especially considering the availability of the MS data in the supplemental files that could be explored by the scientific community. However, the flow chart on Figure 1A still indicates KEGG/GO analysis – please check and revise as appropriate. Aside from this, I have no further comments.

Experimental design

no comment

Validity of the findings

no comment

Reviewer 2 ·

Basic reporting

The authors have greatly improved their manuscript and have addressed all of my concerns. I have no further questions.

Experimental design

The authors have greatly improved their manuscript and have addressed all of my concerns. I have no further questions.

Validity of the findings

The authors have greatly improved their manuscript and have addressed all of my concerns. I have no further questions.

Additional comments

The authors have greatly improved their manuscript and have addressed all of my concerns. I have no further questions.

---

## Round 0.4 · accepted · Accept

Dear Dr. Jia,
Thank you for your submission to PeerJ.
I am writing to inform you that your manuscript - Ropivacaine inhibits the malignant behavior of lung cancer cells by regulating retinoblastoma-binding protein 4 - has been Accepted for publication.

Congratulations!

With kind regards,
Abhishek Tyagi
Academic Editor
PeerJ Life & Environment

Reviewer 1 ·

Basic reporting

The authors have addressed all of my comments.

Experimental design

no comment

Validity of the findings

no comment